# Creutzfeldt–Jakob Disease Mimicking Transient Brain Ischemia in a Patient with a Mitral Valve Prosthesis—A Case Report

**DOI:** 10.3390/reports8040250

**Published:** 2025-11-28

**Authors:** Goda Barauskienė, Medeinė Laurikaitytė, Daiva Emilija Rekienė, Saulius Sadauskas, Albinas Naudžiūnas, Edita Mašanauskienė

**Affiliations:** Internal Medicine Department, Kaunas Hospital, Lithuanian University of Health Sciences, Josvainių g. 2, 47144 Kaunas, Lithuania; medeine.laurikaityte@stud.lsmu.lt (M.L.); daivaemilija.rekiene@lsmu.lt (D.E.R.); saulius.sadauskas@lsmu.lt (S.S.); albinas.naudziunas@lsmu.lt (A.N.); edita.masanauskiene@lsmu.lt (E.M.)

**Keywords:** Creutzfeldt–Jakob disease, prion disease, cardiac prosthesis and dementia, rapidly progressive neurological disorder

## Abstract

**Background and Clinical Significance**: Creutzfeldt–Jakob disease (CJD) is a rare and fatal neurodegenerative disorder caused by prion protein misfolding. The disease poses significant diagnostic challenges, particularly when its initial symptoms mimic other conditions, such as transient ischemic attacks. Early recognition and differentiation from other neurological conditions are critical, as misdiagnosis may lead to unnecessary interventions. This case highlights a unique presentation of CJD in a male Caucasian patient with a history of cardiac surgery and mitral valve prosthesis, emphasizing the role of multidisciplinary evaluation in complex neurological cases. **Case Presentation**: A male patient in his mid-sixties with a history of mitral valve mechanical prosthesis and prior infective endocarditis presented with progressive cognitive decline, memory impairment, and episodes of confusion. Initial cardiovascular investigations suggested mitral valve prosthesis thrombosis, while neurological assessment pointed toward transient brain ischemia. However, brain imaging remained inconclusive. Given the rapid deterioration of cognitive and motor functions, further diagnostic workup was performed. MRI findings revealed cortical diffusion restrictions consistent with probable CJD. Despite symptomatic management, the patient’s condition worsened, leading to akinetic mutism and death within eight days of diagnosis. **Conclusions**: This case underscores the diagnostic complexity of CJD, particularly when initial symptoms overlap with transient ischemic events. It highlights the importance of comprehensive neuroimaging and an interdisciplinary approach in recognizing atypical neurodegenerative diseases to improve diagnostic accuracy and patient management.

## 1. Introduction

Creutzfeldt–Jakob disease, a fatal and uncommon disorder, was named after Dr. Hans Gerhard Creutzfeldt and Dr. Alfons Jakob, who first documented the disease in 1920 and 1921, respectively. This condition is caused by misfolded proteins—prions, which prompt normal proteins in the brain to adopt the same misfolded structure. The disease progresses swiftly, resulting in neurological dysfunction, speech abnormalities, and movement disorders like myoclonus and akinetic mutism. Within six months of onset, over half of patients succumb to the disease, while approximately 80% pass away within a year of its onset [1]. In Lithuania, from 2020 to 2023, only two additional cases were reported [2].

CJD is categorized into four subtypes based on its cause: sporadic (also known as classic form), genetic, iatrogenic, and variant forms [3]. Sporadic Creutzfeldt–Jakob Disease (CJD) is confirmed through standard neuropathological techniques, which include immunocytochemical methods and Western blot analysis to identify protease-resistant PrP (prion protein). A probable diagnosis of sporadic CJD is made when a person exhibits rapidly progressing dementia along with at least two of the following symptoms: myoclonus, visual or cerebellar signs, pyramidal/extrapyramidal signs, or akinetic mutism. Additional supporting evidence for the diagnosis includes a positive outcome in tests such as a typical EEG showing periodic sharp wave complexes, a positive 14-3-3 CSF assay (for patients with a disease duration of less than 2 years), or specific findings on a magnetic resonance imaging (MRI) brain scan [4].

Studying real-life cases is a way to understand such rare diseases as CJD. It helps us gain knowledge about diagnostics, disease development, and finding potential treatments. Despite their relative rarity, prion diseases are an area of increasing research interest due to the recognition among researchers that neurodegenerative disorders like Alzheimer’s and Parkinson’s diseases share a commonality with prion diseases—the aggregation and dissemination of misfolded host proteins. The underlying mechanism bears resemblance to the seeded protein polymerization process observed in prion diseases [5,6,7,8].

## 2. Case Presentation

A male patient in his mid-sixties, with a history of close equine contact, was referred to our neurology department following extensive evaluations in the emergency department and cardiology outpatient clinic due to deterioration of his coordination, spatial awareness, and memory loss. He had a history of mitral valve mechanical prosthesis and tricuspid valve plastic surgery performed 17 years earlier due to infective endocarditis. His medication regimen included spironolactone (25 mg daily), perindopril (5 mg daily), warfarin (5 mg daily), and carvedilol (25 mg daily). A first-degree relative had been diagnosed with Alzheimer’s disease.

The patient, previously asymptomatic, experienced a notable decline in cognitive function two months prior to presentation. He began exhibiting memory disturbances, initially manifesting as transient episodes of disorientation while driving. These episodes, characterized by a loss of spatial awareness, gradually increased in frequency, necessitating constant supervision. Concurrently, he developed symptoms of heart failure, including exertional dyspnea, general weakness, and episodic chest pain.

After experiencing multiple instances of confusion, the patient went to the emergency hospital to seek medical assistance. A cardiac ultrasound was performed, raising a high suspicion of mitral valve prosthesis thrombosis and confirming third-degree aortic valve insufficiency, potentially necessitating surgical intervention. A subsequent brain computed tomography (CT) scan for suspected cerebral ischemia was inconclusive. Neurological assessment revealed only slowed speech, leading to the provisional diagnosis of transient brain ischemia, particularly considering the potential mitral valve prosthesis thrombus. Anticoagulation treatment was revised, and the patient was discharged with instructions to book a cardiologist consultation for valve correction. Regrettably, the patient’s condition rapidly deteriorated at home, characterized by progressive forgetfulness, disorientation, and an inability to perform daily activities. Upon admission, the patient demonstrated rapidly progressive cognitive decline, marked by severe anterograde memory loss, visuospatial disorientation, ideomotor apraxia, bradyphrenia, and mildly dysarthric speech. Neurological examination revealed stimulus-sensitive myoclonus in the upper limbs and mild axial ataxia. He was dependent on assistance for basic daily activities. His spatial awareness deteriorated, causing him to lose track of familiar locations within his home. Eating and other self-care activities became significant challenges, severely impacting his independence.

Despite follow-up consultations in the emergency room and cardiology clinic, which included brain CT and CT angiography scans, as well as neurological and cardiological assessments, a definitive diagnosis remained elusive. Due to a rapidly worsening condition, the patient was hospitalized in the neurology department for further evaluation (Figure 1). The differential diagnostic procedures and their respective diagnoses are outlined in Table 1. A video of the MRI DWI can be viewed here.

A multidisciplinary team was formed to address the complexities of this case. Therapeutic adjustments were implemented for the symptomatic treatment of heart failure, hypertension, and dehydration, while anticoagulation therapy was closely monitored for atrial undulation. Concurrently, a comprehensive diagnostic workup was undertaken.

An infectious disease specialist was consulted and initially suspected Lyme disease, but later considered autoimmune encephalitis due to the atypical symptoms. Although a decision was made to perform a cerebrospinal fluid (CSF) test, it was not carried out due to the rapid progression of the disease and the patient’s anticoagulation treatment.

The diagnostic evaluation comprised an ECG revealing normosystolic atrial undulation, and biochemical assessments confirming normal renal, hepatobiliary, and endocrine functions with no abnormalities in infectious or haematological profiles. Pulmonary tests were unremarkable, a fundus examination was conducted, and HIV testing returned negative. Further cardiovascular reassessment showed no evidence of a thrombus on the mitral valve prosthesis, and given the mismatch with ischemic clinical signs, a repeat CT was performed, but it remained inconclusive. The encephalography revealed disorganized and slowed bioelectric activity across all brain regions, particularly in the centro-temporo-occipital areas (more pronounced on the right), absence of physiological alpha rhythm waves, and generalized epileptiform activity in both hemispheres. Suspecting rare diseases and to explore potential demyelinating processes, an MRI was subsequently conducted. MRI revealed diffusion restriction in the cerebral cortex, primarily affecting the parietal, occipital lobes and cingulate gyrus, with less involvement observed in the fronto-temporal lobe, right caudate nucleus, and anterior putamen. The MRI findings supported the probable diagnosis of Creutzfeldt–Jakob disease (Figure 1).

The initial differential diagnosis considered recurrent cerebral ischemia influenced by the presence of a mitral valve prosthesis and underlying heart disease. However, the rapid progression of dementia and the emergence of symptoms not typical of ischemia necessitated a more extensive evaluation to explore the possibility of less common diseases. A multidisciplinary approach subsequently broadened the diagnostic considerations to include autoimmune or infectious encephalitis, neuroborreliosis, and metabolic disorders. Although initial analyses were inconclusive, the worsening of neurological symptoms prompted the performance of a brain MRI, which disclosed alterations suggestive of Creutzfeldt–Jakob disease (CJD). The combination of clinical observations and MRI findings ultimately led to a provisional diagnosis of CJD. The differential diagnostic procedures and their respective diagnoses are outlined in Figure 1.

Therapeutic adjustments were implemented for the symptomatic treatment of heart failure, hypertension, and dehydration. Anticoagulation therapy was closely monitored for atrial undulation. Despite these efforts, the rapid progression of the disease limited the scope of treatment options.

## 3. Results

The patient experienced rapid neurological deterioration, manifesting ataxia, widespread myoclonus, and eventually progressing to akinetic mutism. Despite comprehensive medical efforts, his condition continued to decline, resulting in his death just eight days after the diagnosis. This case highlights the diagnostic challenges and complexities associated with neurodegenerative disorders, emphasizing the necessity for heightened clinical suspicion and thorough evaluation in atypical presentations.

Despite the seriousness of the diagnosis, the choice was made not to perform an autopsy due to the expensive nature of the procedure and out of consideration for the wishes of the patient’s family. It is important to emphasize the immense emotional burden placed on the family, who were profoundly impacted by the sudden and rapid deterioration of the patient’s health, ultimately resulting in a distressing loss.

## 4. Discussion and Conclusions

The prion protein, referred to as PrP, can adopt two distinct structural conformations: PrP^C, the physiological form present in mammalian cells that participates in normal cellular activities, and PrP^Sc, the pathological isoform. PrP^Sc is characterized by its ability to convert the benign PrP^C into a misfolded state [9]. PrP^C is a widely expressed cell surface glycoprotein that is known to be implicated in intracellular signaling cascades and is most abundantly found in the outer leaflet of the plasma membrane in the central nervous system [10,11]. Research has shown that PrP^C plays a crucial role in the NCAM (neural cell adhesion molecule)-dependent neuronal differentiation of neural stem and precursor cells [12]. Additionally, studies have demonstrated that PrP^C is essential in regulating synaptic plasticity in the developing hippocampus through protein kinase A (PKA), thereby contributing to proper synaptic formation in adulthood [13]. PrP^C is also involved in the formation and maintenance of myelin [11,14], anti-apoptotic roles (during oxidative stress-induced cell death), pro-apoptotic roles (in ER stress), Cu^2+^ binding [15], learning and memory [16], as well as sleep patterns [17].

When PrP^C misfolds, whether as a result of sporadic mutations, genetic factors or ingestion of infectious prion agents, it undergoes a transformation into a protein with a predominantly betasheet-rich conformation. This altered structure confers heightened resistance to the typical protein degradation mechanisms [18]. Named after the Scrapie disease (Ovine and caprine spongiform encephalopathy) in the 1980s, PrP^Sc induces normal PrP^C membrane protein to complete unfolding, followed by refolding through a series of molecular events in which PrP^Sc acts as a physical template [19]. Accumulation and propagation of PrP^Sc in the brain form insoluble plaques or fibrils, which disrupt normal brain function and cause neuron death. The spongiform appearance in Creutzfeldt–Jakob Disease (CJD) refers to the characteristic microscopic appearance of brain tissue affected by the disease, which shows a sponge-like pattern with numerous small holes or vacuoles in the neuropil, a result of neuronal damage and loss [20]. It is worth mentioning that the incubation stage for Creutzfeldt–Jakob disease can last up to several decades. Reviewing past histopathological examinations of tonsil and appendix samples revealed evidence suggesting that thousands of individuals carry PrP^Sc in their bodies, though due to long incubation, they never developed symptoms [21]. In this case, the absence of an autopsy precludes a definitive disease classification. The patient’s extensive animal contact raises the possibility of a variant aetiology, though this remains speculative. Ongoing surveillance is essential to assess the incidence of Creutzfeldt–Jakob disease (CJD) in similar contexts.

In the clinical case under examination, the initial differential diagnosis considered recurrent cerebral ischemia influenced by the presence of a mitral valve prosthesis and underlying heart disease. However, the rapid progression of dementia and the emergence of symptoms not typical of ischemia necessitated a more extensive evaluation to explore the possibility of less common diseases. A multidisciplinary approach subsequently broadened the diagnostic considerations to include autoimmune or infectious encephalitis, neuroborreliosis, and metabolic disorders. Although initial analyses were inconclusive, the worsening of neurological symptoms prompted the performance of a brain MRI, which disclosed alterations suggestive of Creutzfeldt–Jakob disease (CJD). The combination of clinical observations and MRI findings ultimately led to a provisional diagnosis of CJD.

Given the unusually rapid clinical decline, a broad differential workup was performed to exclude other possible causes of rapidly progressive encephalopathy. Infectious, autoimmune, metabolic, and vascular etiologies were ruled out through negative serological tests, normal biochemical and inflammatory markers and unremarkable CT and CT-angiography findings. In the context of typical MRI and EEG features, these results supported a probable diagnosis of CJD.

MRI revealed diffusion restriction in the cerebral cortex, predominantly in the parietal, occipital lobes, and cingulate gyrus, with lesser involvement in the fronto-temporal lobe, right caudate nucleus, and anterior putamen. Additionally, there were signs of previous microhemorrhages in the temporo-parietal areas, indicated by hemosiderin deposits.

The observed pathology in the parietal and occipital lobes correlates clinically with an array of symptoms that evolved over time. Initially, the patient demonstrated spatial disorientation, a symptom often associated with parietal lobe dysfunction due to its role in spatial perception and processing. This was followed by the onset of apraxia and visual agnosia, indicative of impaired visuospatial skills and object recognition, respectively, which are functions primarily governed by the occipital and parietal lobes.

EEG findings of disorganized, slowed activity, absence of alpha rhythm, and generalized epileptiform discharges were highly suggestive of severe and diffuse brain pathology, aligning with the clinical picture of Creutzfeldt–Jakob Disease.

The progression of symptoms and imaging findings was consistent with CJD, highlighting the diagnostic challenges and complexities associated with neurodegenerative disorders. This case underscores the importance of a thorough and multidisciplinary diagnostic approach when dealing with atypical neurological presentations.

## Figures and Tables

**Figure 1 reports-08-00250-f001:**
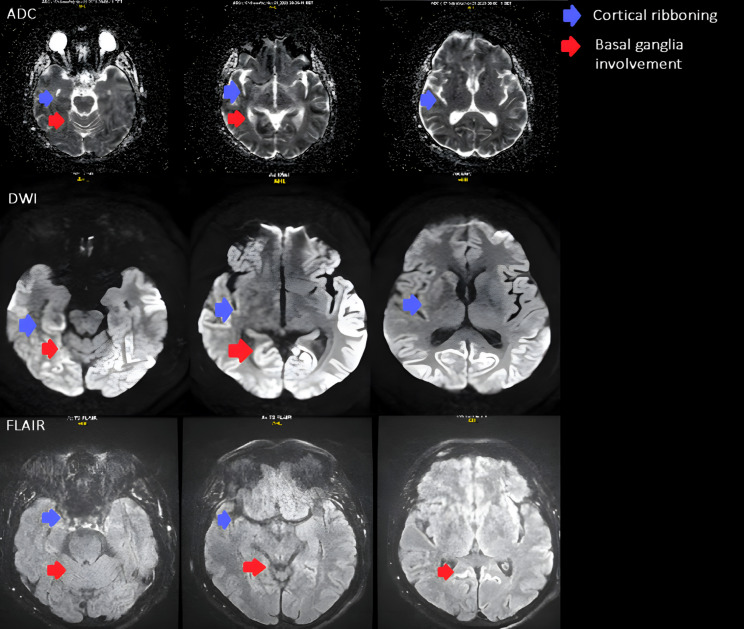
Axial brain MRI shows signal changes typical of Creutzfeldt–Jakob disease. Restricted diffusion is seen in the parietal and occipital cortices (cortical ribboning, blue arrows) and in the basal ganglia, involving the caudate nucleus and anterior putamen (red arrows). The ADC maps demonstrate reduced diffusion, and corresponding hyperintensities on FLAIR confirm the same areas of involvement.

**Table 1 reports-08-00250-t001:** Diagnostic Procedures and Corresponding Differential Diagnoses. This table outlines various diagnostic tests and procedures along with their respective differential diagnoses in the context of encephalopathies and other neurological conditions.

Test/Diagnostic Procedure	Differential Diagnosis
**Blood Glucose**	Hypoglycemic Encephalopathy
**Electrolytes and Renal Function Tests**	Various metabolic encephalopathies, electrolyte disbalance
**Liver Function Tests**	Toxic Encephalopathy, Hepatic Encephalopathy
**CT and CT Angiography**	Cerebral Ischemia, Intracerebral Haemorrhage, and Brain Tumors
**Lyme Disease Immunoglobulins (Serology)**	Lyme Encephalitis (neuroborreliosis)
**EEG (Electroencephalogram)**	Epilepsy
**MRI (Magnetic Resonance Imaging)**	Differentiate from rapidly progressing dementia types, Alzheimer’s Disease, Parkinson-plus syndromes
**HIV Antibody Test**	HIV—associated Neurocognitive Disorder
**CRP, ESR, and Other Inflammatory Markers**	Autoimmune Encephalitis, Viral Encephalitis
**Lumbar Puncture (postponed due to present hypocoagulation)**	Autoimmune Encephalitis, Viral Encephalitis, and Prion Diseases

## Data Availability

The original contributions presented in this study are included in the article. Further inquiries can be directed to the corresponding author.

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
