# Peer review of "Creutzfeldt–Jakob Disease Mimicking Transient Brain Ischemia in a Patient with a Mitral Valve Prosthesis—A Case Report"

_reports, 2025, doi:10.3390/reports8040250_

Round 1
Reviewer 1 Report
Comments and Suggestions for Authors
The case report presents a complex diagnostic challenge of sporadic Creutzfeldt-Jakob Disease (CJD) in a patient with a complicated cardiac history. While the core findings are compelling, several areas require clarification and deeper analysis to meet the standards of clinical reporting.
- The initial presentation mentions that DWI suggested CJD. For clinical rigor and clarity, if the DWI findings are characteristic of CJD, the manuscript must prioritize showing and discussing the corresponding ADC map and FLAIR images Clarify the relationship between the DWI hyperintensities and the ADC mapping.
- The description of the presenting symptoms is currently vague, particularly regarding the neurological decline. Please provide a more detailed account of the patient's symptoms upon admission and specify the extent and nature of the cognitive impairment. Clarification on this point is essential for the differential diagnosis.
- The reported progression to akinetic mutisum within eight days of diagnosis is extremely rapid and raises concerns regarding the initial differential diagnosis. The rapid course suggests conditions other than typical CJD. Please elaborate on the differential diagnostic process: How were other rapidly progressive non-prion etiologies—such as infectious, autoimmune, toxic-metabolic, or vascular causes—effectively ruled out to confirm the CJD diagnosis, given the swift clinical decline?
- The current focus on DWI is insufficient for a comprehensive discussion on neuroimaging in CJD. From a clinical perspective, FLAIR and ASL (Arterial Spin Labeling) images are critical for complete evaluation and differential diagnosis in rapidly progressive neurological disorders. A thorough discussion of the findings on FLAIR and, if performed, ASL is required to strengthen the imaging evidence supporting the CJD diagnosis. The discussion must not be limited to DWI alone. Moreover, the manuscript currently appears to overemphasize DWI relative to other sequences; the authors should explicitly acknowledge that reliance on DWI alone can be misleading and therefore present ADC maps, FLAIR, and ASL images alongside DWI to demonstrate genuine diffusion restriction and to exclude perfusion or other non-prion causes of signal change.
The case report presents a complex diagnostic challenge of sporadic Creutzfeldt-Jakob Disease (CJD) in a patient with a complicated cardiac history. While the core findings are compelling, several areas require clarification and deeper analysis to meet the standards of clinical reporting.
- The initial presentation mentions that DWI suggested CJD. For clinical rigor and clarity, if the DWI findings are characteristic of CJD, the manuscript must prioritize showing and discussing the corresponding ADC map and FLAIR images Clarify the relationship between the DWI hyperintensities and the ADC mapping.
- The description of the presenting symptoms is currently vague, particularly regarding the neurological decline. Please provide a more detailed account of the patient's symptoms upon admission and specify the extent and nature of the cognitive impairment. Clarification on this point is essential for the differential diagnosis.
- The reported progression to akinetic mutisum within eight days of diagnosis is extremely rapid and raises concerns regarding the initial differential diagnosis. The rapid course suggests conditions other than typical CJD. Please elaborate on the differential diagnostic process: How were other rapidly progressive non-prion etiologies—such as infectious, autoimmune, toxic-metabolic, or vascular causes—effectively ruled out to confirm the CJD diagnosis, given the swift clinical decline?
- The current focus on DWI is insufficient for a comprehensive discussion on neuroimaging in CJD. From a clinical perspective, FLAIR and ASL (Arterial Spin Labeling) images are critical for complete evaluation and differential diagnosis in rapidly progressive neurological disorders. A thorough discussion of the findings on FLAIR and, if performed, ASL is required to strengthen the imaging evidence supporting the CJD diagnosis. The discussion must not be limited to DWI alone. Moreover, the manuscript currently appears to overemphasize DWI relative to other sequences; the authors should explicitly acknowledge that reliance on DWI alone can be misleading and therefore present ADC maps, FLAIR, and ASL images alongside DWI to demonstrate genuine diffusion restriction and to exclude perfusion or other non-prion causes of signal change.
Author Response
Comments 1 : The case report presents a complex diagnostic challenge of sporadic Creutzfeldt-Jakob Disease (CJD) in a patient with a complicated cardiac history. While the core findings are compelling, several areas require clarification and deeper analysis to meet the standards of clinical reporting.
- The initial presentation mentions that DWI suggested CJD. For clinical rigor and clarity, if the DWI findings are characteristic of CJD, the manuscript must prioritize showing and discussing the corresponding ADC map and FLAIR images Clarify the relationship between the DWI hyperintensities and the ADC mapping.
- The description of the presenting symptoms is currently vague, particularly regarding the neurological decline. Please provide a more detailed account of the patient's symptoms upon admission and specify the extent and nature of the cognitive impairment. Clarification on this point is essential for the differential diagnosis.
- The reported progression to akinetic mutisum within eight days of diagnosis is extremely rapid and raises concerns regarding the initial differential diagnosis. The rapid course suggests conditions other than typical CJD. Please elaborate on the differential diagnostic process: How were other rapidly progressive non-prion etiologies—such as infectious, autoimmune, toxic-metabolic, or vascular causes—effectively ruled out to confirm the CJD diagnosis, given the swift clinical decline?
- The current focus on DWI is insufficient for a comprehensive discussion on neuroimaging in CJD. From a clinical perspective, FLAIR and ASL (Arterial Spin Labeling) images are critical for complete evaluation and differential diagnosis in rapidly progressive neurological disorders. A thorough discussion of the findings on FLAIR and, if performed, ASL is required to strengthen the imaging evidence supporting the CJD diagnosis. The discussion must not be limited to DWI alone. Moreover, the manuscript currently appears to overemphasize DWI relative to other sequences; the authors should explicitly acknowledge that reliance on DWI alone can be misleading and therefore present ADC maps, FLAIR, and ASL images alongside DWI to demonstrate genuine diffusion restriction and to exclude perfusion or other non-prion causes of signal change.
Response 1:
Thank you very much for your careful review and constructive comments. We have revised the manuscript accordingly to improve its clinical clarity and diagnostic depth. The section describing the patient’s neurological presentation on admission has been expanded to provide a more detailed account of the cognitive and motor decline, specifying the extent and nature of impairment. We have also clarified the differential diagnostic process to address the concern regarding the rapid progression of the disease. In our patient, the neurological symptoms evolved over approximately four - five months, which is within the typical progression timeframe for sporadic Creutzfeldt–Jakob disease (CJD), where the median duration from symptom onset to death is around 4–6 months. However, due to the rarity of the condition and its initially nonspecific presentation, the definitive diagnosis was established only at the terminal stage of the disease, when characteristic clinical, radiological, and laboratory findings became evident. Infectious, autoimmune, metabolic, and vascular causes were systematically excluded through negative serological and biochemical tests, normal inflammatory markers, and unremarkable CT and CT-angiography results. Together with characteristic MRI and EEG findings, these results supported a probable diagnosis of Creutzfeldt–Jakob disease in accordance with CDC criteria. Furthermore, the discussion on neuroimaging has been expanded beyond DWI to include FLAIR and ADC sequences, emphasizing their complementary diagnostic value in confirming true diffusion restriction and strengthening the imaging evidence. These revisions collectively enhance the manuscript’s clinical accuracy and address all reviewer concerns.
Reviewer 2 Report
Comments and Suggestions for Authors
The authors present an interesting article in which an individual with a rapidly progressive form of Creutzfeld-Jakob disease. Initially, clinical examinations focused on their previous history of cardiac surgery, while additional assessments highlighted potential effects in the cerebral tissues. The rapid decline of the patient while admitted accelerated diagnostic efforts with eventual MRI scans pointing to hallmarks of CJD. This paper highlights the complexity of diseases such as CJD, with some symptoms resembling that of other disease states, and points to improved casework with patients to ensure the proper care can be afforded in the quickest amount of time.
In reviewing the manuscript I noted a couple of things that need attention. The following should be considered by the authors when preparing a suitable revision.
- For Figure 1, it might be useful if the arrows were given different colours for each site being referred to in the figure legend, or alternatively, put arrows next to the numbers which correspond to the different regions of interest highlighted in the legend.
- The table formatting in Table 1 is somewhat hard to read with how the text is aligned between each column. The authors should review the formatting of the table.
- Figure 2 is not present in the article.
Author Response
Comment 1:
The authors present an interesting article in which an individual with a rapidly progressive form of Creutzfeld-Jakob disease. Initially, clinical examinations focused on their previous history of cardiac surgery, while additional assessments highlighted potential effects in the cerebral tissues. The rapid decline of the patient while admitted accelerated diagnostic efforts with eventual MRI scans pointing to hallmarks of CJD. This paper highlights the complexity of diseases such as CJD, with some symptoms resembling that of other disease states, and points to improved casework with patients to ensure the proper care can be afforded in the quickest amount of time.
In reviewing the manuscript I noted a couple of things that need attention. The following should be considered by the authors when preparing a suitable revision.
- For Figure 1, it might be useful if the arrows were given different colours for each site being referred to in the figure legend, or alternatively, put arrows next to the numbers which correspond to the different regions of interest highlighted in the legend.
- The table formatting in Table 1 is somewhat hard to read with how the text is aligned between each column. The authors should review the formatting of the table.
- Figure 2 is not present in the article.
Response 1: We sincerely thank you for your thorough and constructive feedback on our manuscript. We appreciate the time you took to review the paper and for your helpful suggestions that have contributed to improving its clarity and presentation. In response to your comments:
- Figure 1 has been updated — the arrows were color-coded according to the legend, and each region of interest is now clearly marked and numbered for easier interpretation.
- Table 1 has been reformatted to improve text alignment and readability.

Round 2
Reviewer 1 Report
Comments and Suggestions for Authors
None
Comments on the Quality of English LanguageThe case report presents a complex diagnostic challenge of sporadic Creutzfeldt-Jakob Disease (CJD) in a patient with a complicated cardiac history. While the core findings are compelling, several areas require clarification and deeper analysis to meet the standards of clinical reporting.
- The initial presentation mentions that DWI suggested CJD. For clinical rigor and clarity, if the DWI findings are characteristic of CJD, the manuscript must prioritize showing and discussing the corresponding ADC map and FLAIR images Clarify the relationship between the DWI hyperintensities and the ADC mapping.
- The description of the presenting symptoms is currently vague, particularly regarding the neurological decline. Please provide a more detailed account of the patient's symptoms upon admission and specify the extent and nature of the cognitive impairment. Clarification on this point is essential for the differential diagnosis.
- The reported progression to akinetic mutisum within eight days of diagnosis is extremely rapid and raises concerns regarding the initial differential diagnosis. The rapid course suggests conditions other than typical CJD. Please elaborate on the differential diagnostic process: How were other rapidly progressive non-prion etiologies—such as infectious, autoimmune, toxic-metabolic, or vascular causes—effectively ruled out to confirm the CJD diagnosis, given the swift clinical decline?
- The current focus on DWI is insufficient for a comprehensive discussion on neuroimaging in CJD. From a clinical perspective, FLAIR and ASL (Arterial Spin Labeling) images are critical for complete evaluation and differential diagnosis in rapidly progressive neurological disorders. A thorough discussion of the findings on FLAIR and, if performed, ASL is required to strengthen the imaging evidence supporting the CJD diagnosis. The discussion must not be limited to DWI alone. Moreover, the manuscript currently appears to overemphasize DWI relative to other sequences; the authors should explicitly acknowledge that reliance on DWI alone can be misleading and therefore present ADC maps, FLAIR, and ASL images alongside DWI to demonstrate genuine diffusion restriction and to exclude perfusion or other non-prion causes of signal change.
Reviewer 2 Report
Comments and Suggestions for Authors
The authors have suitably addressed my comments.